# Designing target trials using electronic health records: A case study of second-line disease-modifying anti-rheumatic drugs and cardiovascular disease outcomes in patients with rheumatoid arthritis

Adovich S. Rivera[1,2], Jacob B. Pierce[3], Arjun Sinha[4], Anna E. Pawlowski[5], Donald M. Lloyd-Jones[4,6], Yvonne C. Lee[7], Matthew J. Feinstein[4,6], Lucia C. Petito[8]*

1 Institute for Public Health and Management, Northwestern University Feinberg School of Medicine, Chicago, Illinois, United States of America, 2 Department of Research and Evaluation, Kaiser Permanente Southern California, Pasadena, California, United States of America, 3 Department of Medicine, Duke University School of Medicine, Durham, North Carolina, United States of America, 4 Department of Medicine, Division of Cardiology, Northwestern University Feinberg School of Medicine, Chicago, Illinois, United States of America, 5 Northwestern Medicine Enterprise Data Warehouse, Northwestern University, Chicago, Illinois, United States of America, 6 Department of Preventive Medicine, Division of Epidemiology, Northwestern University Feinberg School of Medicine, Chicago, Illinois, United States of America, 7 Department of Medicine, Division of Rheumatology, Feinberg School of Medicine, Northwestern University, Chicago, Illinois, United States of America, 8 Department of Preventive Medicine, Division of Biostatistics, Northwestern University Feinberg School of Medicine, Chicago, Illinois, United States of America

* lucia.petito@northwestern.edu

## Abstract

### Background

Emulation of the "target trial" (TT), a hypothetical pragmatic randomized controlled trial (RCT), using observational data can be used to mitigate issues commonly encountered in comparative effectiveness research (CER) when randomized trials are not logistically, ethically, or financially feasible. However, cardiovascular (CV) health research has been slow to adopt TT emulation. Here, we demonstrate the design and analysis of a TT emulation using electronic health records to study the comparative effectiveness of the addition of a disease-modifying anti-rheumatic drug (DMARD) to a regimen of methotrexate on CV events among rheumatoid arthritis (RA) patients.

### Methods

We used data from an electronic medical records-based cohort of RA patients from Northwestern Medicine to emulate the TT. Follow-up began 3 months after initial prescription of MTX (2000–2020) and included all available follow-up through June 30, 2020. Weighted pooled logistic regression was used to estimate differences in CVD risk and survival. Cloning was used to handle immortal time bias and weights to improve baseline and time-varying covariate imbalance.

**Data Availability Statement:** Because of the sensitive nature of the data analyzed for this study,

requests to access the dataset from qualified researchers trained in human subject confidentiality protocols may be sent to MF (matthewjfeinstein@northwestern.edu) or LCP (lucia.petito@northwestern.edu) and the Northwestern Institutional Review board (irb@northwestern.edu). The data contain potentially identifying and sensitive patient information and, therefore, cannot be readily shared in full to protect confidentiality and privacy. Anonymized data that support the findings of this study may be made available from the team after approval by the Northwestern IRB and meeting the following conditions: 1) agreement to collaborate with the study team on all publications, 2) provision of external funding for administrative and investigator time necessary for this collaboration, 3) demonstration that the external investigation team is qualified and has documented evidence of training human subjects protections, and 4) agreement to abide by the terms outlines in data use agreements between institutions.

**Funding:** The author(s) received no specific funding for this work.

**Competing interests:** ASR was supported by the American Heart Association Predoctoral Fellowship (825793) for unrelated research. LCP receives funds for unrelated research from Omron Healthcare Co., Ltd. Other authors have no other conflicts to declare. The mentioned organizations had no role in study design, data collection and analysis, decision to publish, or preparation of the manuscript. This does not alter our adherence to PLOS ONE policies on sharing data and materials

## Results

We identified 659 eligible people with RA with average follow-up of 46 months and 31 MACE events. The month 24 adjusted risk difference for MACE comparing initiation vs non-initiation of a DMARD was -1.47% (95% confidence interval [CI]: -4.74, 1.95%), and the marginal hazard ratio (HR) was 0.72 (95% CI: 0.71, 1.23). In analyses subject to immortal time bias, the HR was 0.62 (95% CI: 0.29–1.44).

## Conclusion

In this sample, we did not observe evidence of differences in risk of MACE, a finding that is compatible with previously published meta-analyses of RCTs. Thoughtful application of the TT framework provides opportunities to conduct CER in observational data. Benchmarking results of observational analyses to previously published RCTs can lend credibility to interpretation.

## Introduction

Comparative effectiveness research (CER) is crucial for developing practice guidelines [1]. Randomized controlled trials (RCTs) are the gold standard evidence in CER, however, RCTs are not always feasible or ethical and have been criticized for their lack of representativeness of the target patient population [2, 3]. As such, researchers have turned to observational data, including electronic health records (EHR), to conduct CER. The target trial (TT) approach has emerged as an important framework for the design and analysis of CER from observational data [4–6]. Several studies have demonstrated that design and emulation of a hypothetical trial (the "target trial") in observational data can provide reliable estimates of causal effects in CER, after alleviating concerns regarding common biases by benchmarking analyses to previously published RCTs [7–9]. Additionally, trial emulations can be conducted in more diverse populations than the original trials, expanding the generalizability of treatment effects to understudied populations [10].

This TT approach has not been widely adopted in cardiovascular health research. A systematic review found only 200 trial emulations published from March 2012 to October 2022 with 25% utilizing EHR data [11]. Among these papers, 30 were classified as cardiology and 19 identified to focus on major cardiovascular events as an outcome. To improve accessibility, researchers have published TT demonstrations tackling various common question types, often focused on interventions initiated at a single specific index event that corresponds to a clinically-relevant decision point or using administrative datasets [12–16]. We contribute to the emerging TT literature by demonstrating trial emulation to assess the effect of initiating a second-line treatment in addition to first-line treatment on health outcomes: the effect of adding a disease-modifying anti-rheumatic drugs (DMARD) to a regimen of methotrexate on cardiovascular disease in patients with rheumatoid arthritis (RA). RCTs to address this question may not be feasible due to low event rates necessitating large samples or longer follow-up and may not be ethical due to lack of equipoise. In this case, the Food and Drug Administration has encouraged the addition of observational CER studies to post-market safety evidence; [17] the methods described here can be generalized to comparisons of therapies that are confounded by treatment due to indication. This approach can also be used in scenarios where treatment can be initiated at multiple time points and serve as a more principled alternative to the

commonly utilized approach of comparing never to ever initiators. Here, we summarize principles in TT emulation using EHR data and provide additional details about design and implementation to supplement existing guides to TT emulation. Additionally, we provide considerations specific to this research question with the hope that readers will consider this guide when applying the TT approach to their own work.

### Motivating example: Second-line DMARD therapy versus methotrexate monotherapy to reduce cardiovascular events in RA patients

RA is a chronic inflammatory disease characterized by broad activation of the innate and adaptive immune systems [18]. Due to immune activation, people with RA have increased risk of cardiovascular disease (CVD) [19–21]. New biologic and targeted synthetic DMARDs are efficacious in addressing symptoms when methotrexate (MTX) monotherapy has been insufficient [22]. However, the effects of the adding DMARDs to MTX on CVD risk are uncertain. Meta-analyses including only RCTs concluded that the addition of DMARDs did not reduce CVD risk in RA patients, while another meta-analysis that included both RCTs and observational studies suggested that adding DMARDs provided some benefit [23, 24]. The discrepancy may be attributed to previously detailed issues with observational studies such as selection bias, immortal time bias, and unmeasured confounding [4, 9, 25, 26] Here, to address issues with observational studies, we used electronic health record (EHR) data from a large regional academic health system to emulate a (hypothetical) open-label pragmatic trial comparing MTX alone to MTX plus DMARD therapy to assess their effect on CVD risk in RA patients.

## Materials and methods

### Specifying the target trial

The first step in TT emulation is to design the "target trial:" a hypothetical pragmatic RCT designed to assess the effect of an intervention on the outcome(s). The second step is to identify an observational data source, here EHRs, and emulate the TT by analyzing that data [27]. The design process is iterative: key components are described at the beginning and may need to be revisited based on artifacts in the observational data source. Collaboration with clinicians or domain experts is essential in trial emulation, ensuring that analytic decisions do not lead to implausible clinical situations. To aid researchers as they apply this approach to their work, we have included key considerations for the design of each component (Table 1). An overview of the TT and corresponding emulation for our case study is described in Table 2.

### Selecting a data source

In RCTs, recruitment of participants is often done in partnership with health providers or organizations that frequently interact with the target population. In emulation, recruitment is not conducted. Rather, one utilizes found data sources to create a large, prospective cohort of eligible patients. The data source should have reasonable quality and size, so that sufficient variation in treatment strategies is available, and outcome events are prevalent enough. EHRs can be good data sources for clinical outcomes provided that reliable diagnostic algorithms exist and no major changes in data capture occurred. EHRs, however, have inherent issues like irregular timing of visits and informed presence bias which need to be accounted for in the design and reporting of the results.

 For this case study, we created the de-identified and anonymized EHR data from the Northwestern Medicine Enterprise Data Warehouse (NMEDW; Northwestern University Clinical

**Table 1. Considerations when designing a target trial using electronic health records.**

| Protocol Component | Considerations |
|---|---|
| Eligibility criteria | • How do choices around study eligibility translate to the broader population? Do they limit generalizability and transportability?<br>• Is it possible to identify all relevant eligibility criteria from structured data in the EHR? Or are more computationally intensive methods needed to identify information recorded only in physician notes?<br>• What are implications for missing data? Will substantial selection bias be introduced if a complete case analysis is conducted? |
| Treatment strategies | • Are all treatment strategies possible for all types of patients in the eligible population?<br>• Are all candidate treatment strategies used within the eligible population? Is implementation feasible in clinical practice?<br>• If comparing drug classes containing multiple drugs instead individual drugs: do all candidate drugs reasonably have the same expected effect on the outcome? |
| Assignment procedures | • Randomization is assumed conditional on observed confounders (factors that are associated both with treatment strategy decisions and the outcomes of interest). Are all necessary confounders captured in structured data in the EHR? |
| Follow-up period | • How will you identify study baseline (time zero) for all participants? Is there a point at which treatment decisions are often made?<br>• Will all participants begin their treatment simultaneously in time? If not, add a clinically-plausible *grace period* in which treatment initiation is allowable (e.g. initiate a DMARD within 6 months of beginning methotrexate).<br>• How will you identify loss to follow-up in your data source? How will you define contact with the healthcare system? |
| Outcome | • Are sufficient data (e.g., sample size and person-time) available to capture this outcome?<br>• Measurement error: Are operational definitions sensitive or specific enough? Do they capture real-life events of interest?<br>• Do outcomes pass validation with chart review? |
| Causal contrasts of interest | • Do the contrasts of interest of answer questions of significance in relation to clinical practice or policy? |
| Analysis plan | • What is the statistical analysis that would have been conducted for a pragmatic RCT?<br>• Will a validation study be conducted for identifying elements from the EHR? How will you handle missing data?<br>• How will you address confounding and selection (immortal time) bias? |

and Translational Sciences Institute, Chicago, IL, USA), which houses comprehensive outpatient and inpatient EHR data for a large urban health care system in Chicago, Illinois (pull date: June 21 to July 16, 2020). The Northwestern University Institutional review board exempted this study from review and waived informed consent requirements because the research involves the study of deidentified data.

## Eligibility criteria

The eligible population should reflect the population which will be affected by the implications of the research. Only patients who are eligible to receive either treatment strategy should be included; patients with counterindications should be excluded. One may start from criteria in ongoing or completed trials investigating the intervention of interest then adapt these during emulation. Demographic exclusions should be reviewed especially if the goal is to include understudied but often excluded populations. Operational definitions should consider available data. Diagnostic tests may not be routine and might have to be removed as a criterion–or the data source may have to be abandoned or expanded to achieve a sufficient sample size. For EHR analysis, published or validated phenotypes should be used as much as possible [29].

In our ideal TT, RA diagnoses would be confirmed by trained clinicians. This is not feasible with EHR so instead we utilized Internal Classifications of Diseases (ICD) codes, which assumes adequate sensitivity and specificity (S1 Table). We captured newly diagnosed RA

**Table 2. Specification of target trial protocol and emulation in northwestern medicine's enterprise data warehouse (NMEDW).**

| Protocol Component | Description of Target Trial Protocol | Description of Target Trial Emulation Using NMEDW |
|---|---|---|
| Eligibility criteria | Diagnosis of rheumatoid arthritis in patients aged 18-75y between 1/1/2000 and 12/31/2019Management of RA symptoms via methotrexate monotherapy ($\geq$12.5mg/wk) for at least 2 months (8 weeks) prior to time zeroLaboratory Assessment:<br>• estimated glomerular filtration rate (eGFR) >60 mL/min<br>• White blood cell count>3,000/mm$^3$<br>• Absolute neutrophil count>1200/mm$^3$<br>• Liver transaminases<1.5x upper limit of normal<br>• Hemoglobin>9 g/dL<br>• Hematocrit>30%<br>Physician confirmation of no prior history of serious cardiovascular disease including myocardial infarction, heart failure, or coronary revascularization; other autoimmune rheumatic disease (psoriasis, systemic lupus erythematosus, systemic sclerosis, dermatomyositis, polymyositis, atopic dermatitis); inflammatory bowel disease (Crohns, Ulcerative colitis); serious infections (HIV, Hepatitis B, Hepatitis C, Tuberculosis) or cancer excluding nonmelanoma skin cancer prior to time zero. | Same as target trial, except:<br>Lab values can be satisfied using bloodwork taken up to 6 months prior to and 3 months after enrollment<br>Diagnoses will be identified using validated ICD-based definitions instead of physician confirmation |
| Treatment strategies | 1. Initiate second line therapy with any DMARD within 24 months of time zero<br>2. Do not initiate second line DMARD therapy (methotrexate monotherapy)<br>Under both strategies, the decision to discontinue methotrexate or DMARD therapies or initiate any additional therapies is left to the patient and clinician's discretion. | Same, except therapy initiation will be identified through prescription orders. |
| Assignment procedures | Open-label (unblinded) randomization to one treatment strategy at baseline. Participants and clinicians were aware of assigned strategy | Randomization will be assumed conditional on baseline covariates: age, gender, race and ethnicity, insurance, diabetes status, hypertension status, other comorbidity status ($\geq$1 of the following: atrial fibrillation, atherosclerotic CVD, chronic kidney disease, chronic obstructive pulmonary disease), cholesterol level, and eGFR. |
| Follow-up period | Starts at time zero (point of randomization and assignment to treatment strategy) and ends at the earliest of outcome, loss to follow-up, last day of available data (June 30, 2020), or 5 years after time zero | Same, except loss to follow-up is defined as 2 years without a patient encounter at Northwestern. |
| Outcomes | 4-point major adverse cardiovascular event composite:<br>  - Non-fatal MI<br>  - Non-fatal stroke (including hemorrhagic stroke)<br>  - Incident HF (including first hospitalization and outpatient diagnosis)<br>  - Cardiovascular death, certified by a clinician | Same, except components of the outcome will be identified using validated ICD-9 and ICD-10 definitions |
| Causal contrasts of interest | ITT effect; per-protocol effect | Per-protocol effect only<br>Our strategies required the initiation of an DMARD within the grace period regardless of further continuation. This "per-protocol" effect mimics the ITT effect in a trial where all patients initiated their assigned treatment during the grace period. ITT cannot be estimated due to lack of randomization. |
| Analysis plan | ITT analysis; Per-protocol analysis: inverse probability weighted pooled logistic regression model with censoring when participants deviate from study protocol. Weights estimated as a function of baseline (above) and post-baseline covariates: diabetes status, hypertension status, other comorbidity status, cholesterol level, and eGFR | Same, except analysis will be performed in an expanded dataset with 2 replicates (one per treatment strategy) per patient to avoid immortal time bias [28] |

patients to ensure that we had their complete treatment history; this strategy gives us confidence that we captured new second-line treatment users, not referrals of more advanced cases.

## Treatment strategies

RCTs have a large degree of control on the mode, dose, and timing of interventions. For example, trials specify minimum doses for the DMARD to be added or might focus on just one

DMARD [30]. In EHR data, however, there is more treatment variation especially if guidelines do not explicitly favor specific drug(s). The choice of treatment strategies is limited by what is being done in real-world clinical practice; as such, TT emulation is not useful for very new interventions or treatments that are nearly always given for an indication [31].

The definition of the intervention should closely match actions that could be implemented in the real world, so mechanisms for altering the action are concrete. For example, instead of achieving certain blood concentration, we recommend specifying an intervention in terms of the dose of a prescribed drug.

In our analysis, practice guidelines for RA state to use any DMARD as an additional treatment to MTX [22]. Thus, we compare individuals who received MTX monotherapy with those who received any additional DMARD. This simplification bars us from doing head-to-head comparisons of specific DMARDs but captures implications of the guidelines.

## Specifying a grace period

Unlike RCTs where points of randomization are clear and specified in advance, individuals in EHRs do not necessarily share the same timing of treatment initiation/discontinuation. Comparing never to ever DMARD users based on the full data without specifying timing of initiation, induces selection and leads to an unclear causal question. However, using a too strict definitions such as "started additional DMARD exactly after 8 weeks of MTX use" would lead to incredibly small sample sizes. It is also unrealistic because DMARD initiation being off one or two days might be an artifact of data entry, not representing a medical care choice.

One solution is to re-define the intervention to include a grace period–a period of time wherein eligible patients have the option of initiating a treatment strategy. This is a common feature of pragmatic trials that can be emulated in EHRs [32]. The grace period illustrates a tradeoff: the protocol specification is more relaxed, but one captures more individuals and better mimics real-world practices. Grace periods should be realistic and alternative definitions should be included in sensitivity analysis.

Our TT compares individuals who did versus did not receive a DMARD as second-line treatment after MTX within a grace period. In our emulation, eligibility criteria include a minimum MTX treatment duration– 8 weeks–after which participants become eligible to initiate a DMARD. At this point, eligible participants are granted a grace period– 24 months–within which they either initiate an additional DMARD (active treatment) or not (control). This protocol is like a trial where recruitment is not limited to people who were newly diagnosed but instead to people who have at least an 8 week but no more than 2-year history of using MTX alone. This choice accommodates differences in RA disease progression, wherein patients may not need an additional DMARD until clinically indicated. Additionally, our protocol is quite flexible; third- and fourth-line DMARDs are permitted to be initiated anytime after the initial second-line DMARD, and even those in the MTX monotherapy group are considered compliant with protocol if they initiate a DMARDs after the grace period; this is in accordance with the intention-to-treat principle. Stricter protocols can be emulated, but their clinical relevance should be scrutinized.

## Assignment procedures

Treatment assignment in RCTs relies on randomization, which enables unbiased estimation of intention-to-treat (causal) effects [27]. The assignment procedure for any TT must be a pragmatic design wherein patients and providers are aware of the treatment strategy to which they are assigned, as we can never hope to emulate a tightly-controlled, blinded RCT in observational data [26].

For the analysis in EHR data to emulate estimates from the TT, we must try to achieve randomization conditional on *measured* confounders; this conditional randomization is essential to the plausibility of exchangeability of participants between treatment groups. Several strategies have been developed to achieve this goal including propensity score matching, stratification, g-computation (or standardization), and inverse probability weighting [27]. Doubly-robust methods can also be used, although these tend to be computationally intensive [33]. Regardless of chosen statistical approach, one needs to select covariates that act as confounders of the treatment effect: they must be factors measured at or before baseline that influence the treatment assignment decision and are associated with the outcome. (Table 2 for list of covariates in our emulation)

An issue that arises when using EHRs for trial emulation is timing of confounder definitions: the time periods when these confounders are defined matters to maintain temporality. For example, laboratory values that are considered confounders must be assessed prior to treatment assignment, and ideally will have been carried forward for a minimal amount of time (e.g., specifying a look-back window of 12 months, not 10 years). Additionally, one must be careful about informative missingness. For example, total cholesterol might be predictive of being on statins, however, those that are on statins get their lipids measured more frequently. Aside from working with clinicians to capture care practices, careful examination of missingness patterns can help identify these problematic variables.

## Outcomes

Follow-up duration for outcomes, as would be the case for RCTs, should be long enough to capture outcomes of interest, but not so long that biological plausibility is tenuous. The primary outcome of our TT would be occurrence of Major Adverse Cardiac Events (MACE), defined as a 4-point composite CVD outcome including non-fatal myocardial infarction (MI), non-fatal stroke, incident heart failure (HF), and cardiovascular death, adjudicated by clinicians. This outcome is assessed throughout the follow-up period, as defined above.

In our emulation in EHRs, non-fatal MI, non-fatal stroke, and incident HF were identified using validated sets of ICD-9 and ICD-10 codes (S1 Table). As we did not have cause of death recorded in the EHR, we instead used death from all causes in our definition of MACE [21].

## Follow-up period

RCTs have very strict protocols that clearly define a patient's time of enrollment in the study, as well as their time of exit from the study. In the TT framework, "time zero" is the point in time when an individual meets eligibility criteria, treatment is assigned, and follow-up begins; time zero is the observational analog to the date of first treatment received in an RCT. Careful selection of time zero is important to avoid conflating pre- and post-treatment initiation variables, which can lead to immortal time bias [4]. (S1 Fig)

The definition of time zero varies with the clinical research question. It can be met at a single time, for example when studying the effect of initiating remdesivir immediately upon admission and testing positive for COVID-19 affects outcomes [34, 35]. But, it is more common for eligibility to be met at multiple time points, for instance when studying hormone therapy initiation in menopausal women, patients may be continuously eligible throughout menopause. In this setting, a series of sequentially nested trials would need to be conducted [7].

As in an RCT, follow-up ends at the earliest of experiencing a study outcome or loss to follow-up. Loss to follow-up in EHR studies needs to include a measure of inactivity or disenrollment in the healthcare system, as lack of participation precludes us from collecting post-

treatment data. If dates of disenrollment are not available, we recommend pre-specifying a length of time (e.g., 2 years) wherein no contact with the system is considered loss to follow-up.

## Causal contrasts of interest

RCTs often estimate both intention-to-treat (ITT) and per-protocol effects [28, 36]. ITT effects are estimated based on treatment assignment alone, and ignore adherence to treatment protocols. In point-treatment settings at controlled facilities (e.g., a single-dose vaccine trial), all individuals are completely adherent to the protocol, so the ITT effect equals the per protocol effect. However, when the treatment happens over time (e.g., take a medication daily for 3 months), there is no guarantee of perfect adherence so the ITT will not necessarily reflect per-protocol effect. Per-protocol effect estimation accounts for post-treatment assignment protocol adherence, while appropriately adjusting for time-varying confounding (e.g., side effects).

As observational studies do not randomize treatment, we can only estimate per-protocol effects when emulating TTs in EHR. However, we can be less strict in our definition of "protocol." For example, here we attempted to estimate the observational analog of an ITT effect by specifying a protocol that assigned individuals to treatment arms once they initiated a DMARD, and allowed them to change their treatment however they and their physician deemed fit after that initial prescription. Other examples of protocols we could have specified (but did not implement here) are: requiring individuals to refill their prescriptions on a particular schedule, or requiring that they not initiate any other RA treatments before the end of their DMARD prescription.

## Statistical analysis

Once the TT protocol has been defined, an appropriate statistical analysis plan can be developed to address the question of interest. Estimation of ITT and per-protocol effects for survival outcomes in RCTs with non-adherence have been described in detail elsewhere [28]. ITT effects can be estimated using inverse probability of treatment weighted (IPTW) survival models, or baseline-covariate adjusted survival models that are standardized to the empirical baseline covariate distribution. Per-protocol effects are slightly trickier and involve:

1. Estimating time-varying inverse probability of adherence weights [28].

2. Estimating IPTWs using a logistic regression model with treatment as the outcome and baseline covariates as the predictors

3. Using a weighted pooled logistic regression model, where weights are product of those estimated in Step 1 and Step 2. Alternately, using a weighted pooled logistic regression model adjusted for baseline covariates, where weights are estimated in step 1.

The resulting model (Step 3) can then be used to calculate marginal survival curves, risk differences at select times, 5-year mean restricted survival time, and the average hazards ratio over follow-up. Covariates selected to be included in the various models should be guided by existing knowledge or theory and by constructed directed acyclic graphs [37–39]. Researchers may opt to use data-driven approaches for covariate selection (e.g., lasso) but it can add to the computational time and complexity. Covariates can be used in both weighting (steps 1 and 2) and outcome models (step 3) as it may safeguard against residual imbalance [40]. All weights should be stabilized (and possibly truncated) to prevent large weights on rare individuals. Non-parametric bootstrapping can be used to calculate $(1-\alpha)$% confidence intervals [27, 28]. In our emulation, we used a baseline-adjusted weighted pooled logistic regression model, standardized to the empirical distribution of baseline covariates to calculate all marginal effects.

The statistical analysis for our emulation in EHRs should resemble the analysis for the per-protocol effect described above. However, we must address artificial introduction of immortal time bias due to specification of a treatment grace period. Individuals may exhibit behavior consistent with both strategies of interest during the grace period. (S1 Fig) Assigning all individuals who are lost to follow-up or experience an event during the grace period without having initiated a DMARD to the MTX monotherapy arm would artificially inflate the event rate in this arm [41]. There are two possible solutions to avoid this bias:

1. For all individuals who are censored or experience the outcome during the grace period prior to initiating a DMARD, randomly assign them to a treatment strategy. No need for inverse probability of adherence weights.

2. Clone all individuals at study baseline. Assign Clone A to MTX monotherapy, and Clone B to initiate a DMARD within 24 months. Censor individuals when they become non-adherent to their assigned treatment strategy. Use inverse probability of adherence weights to balance time-varying characteristics. Standard errors can be estimated via a non-parametric bootstrap, or in cases with an extreme amount of data, robust variance estimation procedures.

After implementing one of these strategies, estimate the observational analog of an ITT effect via steps described above.

In emulations where cloning or grace periods are not employed, an analysis that can produce conditional exchangeability such as a weighted logistic regression with weights derived from IPW would be sufficient [27]. Matching could also be explored, although care should be undertaken when generalizing findings back to the target samples.

For our emulation, we also conducted sensitivity analyses to explore the impact changing the functional form of time (linear vs quadratic) instead of non-linear splines (main analysis) and the impact of changing the grace period to 12 months instead of 24 months. We also conducted sub-group analysis that included only patients who were diagnosed with RA at least 6 months before time zero. Finally, we conducted a sensitivity analysis where we excluded hydroxychloroquine (HCQ) as a DMARD option. Exclusion of HCQ was done to emulate some previously conducted RCTs where HCQ was allowed as a concurrent therapy to MTX but was not counted as a step-up DMARD [42].

**Missing data.**   Missing data is common in EHRs and can be informative: laboratory values are often only ordered for symptomatic patients. Imputing data can be an effective strategy to mitigate selection bias induced by complete case analyses [43, 44]. Imputation is recommended for all variables included in sample selection (eligibility criteria), baseline covariates, and study outcomes, but not treatment to preserve the integrity of treatment ascertainment. Carry-forward imputation is commonly used despite reservations in the statistical community [45], but in target trials this method is easily feasible. Maximum time should be informed by clinical knowledge; for example, blood pressure changes quickly so should be used proximally to the index date, while lipid levels change slowly so can be carried forward longer.

To limit the computational intensiveness of our TT emulation, we chose to use single imputation for missing baseline variables and carried last observations forward for 2 years for time-varying covariates. All analyses were conducted in R v4.1.0 (see S2 Fig for data observability and S1 File Section for sample code).

## Results

Our final analytic sample consisted of 659 eligible patients with 30,128 person-months of follow-up. (S3 Fig) At baseline, participants were mostly female with a mean age of 54.17 years

(standard deviation (SD): 12.95). Most were White, Non-Hispanic (59.5%) and entered the study in 2014 (SD: 4y). Comorbid conditions were common: 23.4% had HTN, 5.5% had DM, and 5.5% had at least one other comorbidity. (Table 3)

There were 289 (43.6%) patients who initiated second-line DMARD therapy during the grace period (MTX+DMARD), on average 7.6 (SD: 6.68) months after time zero. The three most common DMARDs were adalimumab (71, 25%), hydroxychloroquine (62, 22%), and etanercept (45, 16%). (S2 Table). Among those on MTX monotherapy, 77/370 (20.8%) started a DMARD after the grace period. At the month 24 (end of grace period and point where exposure assignment was finalized), those in the MTX+DMARD group (n = 287) were younger, had a lower proportion of White patients, a higher proportion of Hispanic patients, had a higher proportion with DM and HTN, and had higher eGFR compared to those in the MTX monotherapy (n = 352). (Table 3)

Thirty-one patients (4 from deaths) experienced MACE during the 60-month follow-up, with 20 events occurring during the grace period. The adjusted estimated 60-month MACE-free survival were 90.6% for the MTX+DMARD arm and 89.1% for MTX monotherapy translating to a risk difference of -1.47% (95% CI: -4.74 to 1.95%) and restricted mean survival time (RMST) of 0.57 (95% CI: -0.75 to 1.81) months. The marginal hazard ratio (HR) was 0.72 (95% CI: 0.71 to 1.23) after adjustment for baseline covariates. Results from sensitivity analyses, including altering the functional form of time, restricting the analysis to those diagnosed ≥6 months before time zero, and excluding hydroxychloroquine as a DMARD, did not materially change the conclusions. (Table 4 and Fig 1).

For comparison, we conducted a naïve analysis subject to immortal time bias, wherein eligible patients were retrospectively assigned to treatment based on their available data before 24 months follow-up (S4 Fig). Using a Cox proportional-hazards model adjusted for baseline covariates, the HR for the treatment effect was 0.62 (95% CI: 0.29 to 1.44). As hypothesized, this analysis resulted in an estimated HR that was further from the null; this is probably an artifact of selection bias.

## Discussion

Target trial emulation addresses common issues encountered in analyses of observational data. In this work, we described and applied the TT approach when analyzing EHR data from a large, urban healthcare system (Table 2). Here, we did not observe evidence of differences in MACE risk, a finding that is limited by the small number of outcomes observed. This finding was robust to design-based and statistical choices such as grace period length and functional form of time to specify baseline hazard. Our results contrast with other observational studies that suggested a 30–50% reduction in CVD events with DMARD use [23, 46], but better align with prior meta-analyses that only included RCTs, which found no effect of additional DMARDs on CVD risk.[23, 24] Our work demonstrates that TT emulation with EHR is a feasible approach to conduct CER [26]. Similar to other TT emulation studies, we reproduced results consistent with RCTs using observational data [7, 47].

A key strength of the TT approach is that it requires researchers to state assumptions that affect internal and external validity. This exercise facilitates a systematic approach to study design, principled formulation of an analysis plan, transparent interpretation of results, and collaboration within the research team. In this specific case assessing DMARD addition, the TT approach enabled proper handling of both confounding by indication and avoided immortal time bias. Specifically, we were able to overcome selection bias due to differentially selecting individuals into treatment groups based on post-baseline events. Oftentimes, researchers would do a naïve analysis which compares ever versus never treated (e.g., ever used DMARD

**Table 3. Demographic and clinical characteristics of included patients with rheumatoid arthritis at baseline and stratified by treatment strategy after 24 months, northwestern medicine, January 2000–June 2020.**

| | Baseline | After 24 months | |
| --- | --- | --- | --- |
| | Overall (n = 659) | Addition of Second-Line DMARD Therapy during grace period (n = 287)^ | MTX monotherapy during grace period (n = 352)^ |
| Age at time zero, mean (SD) | 54.17 (12.95) | 52.37 (13.31) | 55.27 (12.54) |
| Male gender, n (%) | 172 (26.1) | 60 (20.9) | 102 (29.0) |
| Race and ethnicity, n (%) | | | |
| Black, non-Hispanic | 92 (14.0) | 40 (13.9) | 46 (13.1) |
| Hispanic | 79 (12.0) | 47 (16.4) | 31 (8.8) |
| White, non-Hispanic | 393 (59.6) | 162 (56.4) | 220 (62.5) |
| Other* | 95 (14.4) | 38 (13.2) | 55 (15.6) |
| Insurance status, n (%) | | | |
| Government | 248 (37.6) | 108 (37.6) | 128 (36.4) |
| Private | 309 (46.9) | 137 (47.7) | 167 (47.4) |
| Uninsured or other | 102 (15.5) | 42 (14.6) | 57 (16.2) |
| Year of Time Zero, mean (SD) | 2014 (4) | 2014 (4) | 2014 (4) |
| Clinical variables+ | | | |
| Hypertension, n (%) | 150 (22.8) | 87 (30.3) | 76 (21.6) |
| Diabetes mellitus, n (%) | 36 (5.5) | 26 (9.1) | 17 (4.8) |
| Other comorbidities, n (%)† | 36 (5.5) | 25 (8.7) | 30 (8.5) |
| eGFR, mean (SD) | 110.49 (36.04) | 88.65 (20.51) | 86.04 (21.84) |
| Total Cholesterol, mean (SD) | 184.48 (20.45) | 182.90 (25.22) | 184.37 (24.97) |

\* - Includes Asian, multiracial, and declined or missing.

† - Includes chronic kidney disease, atrial fibrillation, chronic obstructive pulmonary disease, and atherosclerotic cardiovascular disease. + - Clinical variables were reported at baseline for the overall sample, and updated to reflect most recent values at or prior to 24 months as appropriate. ^ - Difference between baseline and month 24 sample is due to occurrence of outcome or censoring events.

vs never used DMARD) by using future events to assign exposure status. This approach forces the researcher to make "a gamble in which the investigators bet that the amount of selection bias introduced is less than the amount of confounding eliminated."[7] Based on our analysis, the selection induced by doing a naïve comparison might not have altered the overall conclusion. Still, the point estimate of the naïve analysis is further away from the null compared to the emulated value (naïve HR: 0.62 vs emulated OR: 0.71). Importantly, there is a more fundamental issue with the naïve analysis. The comparison answers an unclear question that cannot be applied in real life: How can one ensure a person never uses a drug? How can one ask a person to initiate a drug but not specify when to initiate it? Our paper illustrates an alternative to this problematic naïve approach through design (e.g., introducing grace periods) and analysis (e.g., cloning and re-weighting).

We again stress that the TT emulation framework is not prescriptive in terms of statistical estimation. Depending on the question and data, even the commonly used linear regression model with covariate adjustment may suffice. For our question and data, we used grace periods with weighted pooled logistic regression. Pooled logistic regression has been shown perform comparably with time-dependent Cox models especially in rare outcomes [48]. Use of weights to account for post-baseline confounding have been shown to obtain unbiased estimates in simulation of trial data with null effects [49].We could have also used approaches like marginal

**Table 4. Hazard ratios, risk differences, and restricted mean survival times for 5-year risk of MACE comparing methotrexate monotherapy and addition of second-line DMARD therapy, northwestern medicine, January 2000–June 2020.**

| Analysis† | Marginal HR* | Risk Difference at month 60* | RMST at month 60* |
|---|---|---|---|
| Main | 0.717 (0.709 1.228) | -1.47 (-4.74, 1.95) | 0.573 (-0.751, 1.807) |
| *Sensitivity analyses* | | | |
| 12-month grace period | 0.723 (0.537, 1.270) | -2.1 (-6.86, 2.54) | 0.778 (-0.945, 2.508) |
| Linear time | 1.066 (0.208, 1.123) | -0.9 (-4.49, 2.38) | 0.351 (-0.884, 1.754) |
| Square time | 0.711 (0.241, 1.107) | -1.35 (-4.64, 2.04) | 0.529 (-0.773, 1.820) |
| Diagnosis of RA at least 6 months before time zero | 0.880 (0.744, 1.330) | -0.32 (-4.14, 3.39) | 0.120 (-1.238, 1.594) |
| HCQ excluded from DMARD | 1.031 (0.695, 1.345) | -0.76 (-4.45, 3.65) | 0.262 (-1.138, 1.517) |

HCQ–hydroxychloroquine.

* - 95% percentile bootstrap confidence intervals (CI). Weights and outcomes models adjust for age, gender, race and ethnicity, diabetes, hypertension, and other comorbidity status, baseline cholesterol level, and baseline eGFR.

† - Model used for the denominator of the weights calculation included baseline and time-varying treatment status, comorbidity status, and laboratory values.

structural modelling, longitudinal matching or sequential nested trials to overcome the challenges of time-varying exposures although it would necessitate modifying the question being answered. (e.g., matching produces average treatment among the treated) [50, 51]. Despite key strengths, our emulation has several limitations. First, our data only consisted of structured data–ICD codes and prescription data–from a single health system. As we were unable to include clinical assessments (e.g., pain scores and function assessments) and markers of inflammation in our models [22], our analyses may be subject to unmeasured confounding. Moreover, people may receive care from other facilities and that external data (e.g., prior MTX prescriptions, state death registry) may not be recorded correctly in the NMEDW, so measurement error may have affected study eligibility, treatment identification, and outcome ascertainment. Second, we used a logistic regression model for our weights and outcomes model with some parametric form assumptions. Recent work on causal inference has argued for incorporating more flexible methods like machine learning models; [52] however, in our case integrating these methods would be computationally expensive for little gain. Third, our definition of MACE used all-cause death instead of CVD-specific death. While this definition is consistent with some other studies, these results may not be directly comparable to those from studies whose MACE definition included CVD-specific death [53]. Finally, we were unable to examine individual DMARDs separately due to sample size limitations. This choice implies that each DMARD affects CVD risk equally, which may not be true as conventional and targeted DMARDs operate via different hypothesized mechanisms [54]. A larger dataset with greater treatment heterogeneity is required to investigate DMARD-specific effects on CVD risk.

We designed and emulated a target trial in EHR data from one health system to study the comparative effectiveness of second-line DMARD therapy versus methotrexate monotherapy on CVD risk in RA patients. Our results are limited by sample size, namely number of MACE events observed, although our estimates are compatible with those estimated in meta-analyses

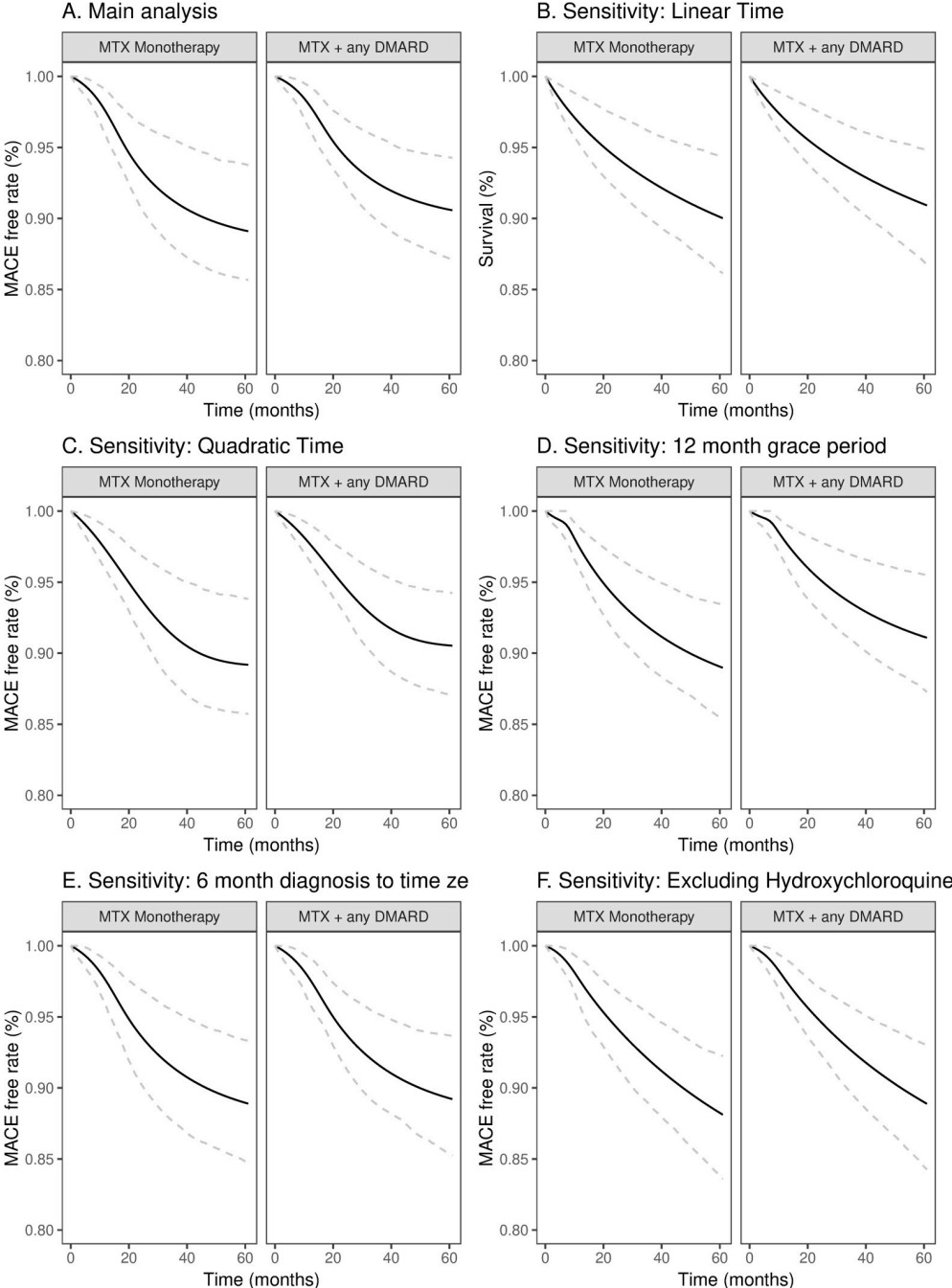

**Fig 1. MACE-free survival curves comparing methotrexate monotherapy versus addition of second-line DMARD therapy, northwestern medicine, January 2000–June 2020.** Caption: Black lines represent survival curves. Dashed gray lines represent 2.5 and 97.5 bootstrapped percentiles from 500 re-samples. Sensitivity analyses included: (B) linear time, (C) linear and quadratic time, (D) used 12-month grace period, (E) require diagnosis of RA≥6 months before time zero, (F) exclude individuals who used hydroxychloroquine as additional therapy.

of RCTs. While RCTs remain the gold standard for evidence in making clinical decisions and practice guidelines, studies that thoughtfully apply the TT framework and benchmark results to prior RCTs provide opportunities to do rigorous CER in observational data.

## Supporting information

**S1 Fig. Illustration of immortal time bias in target trial emulation with a grace period.**
Abbreviations: DMARD–disease-modifying antirheumatic drug, MTX–methotrexate. This
figure illustrates data from 4 hypothetical participants. Yellow represents time available in data
prior to time zero (not included in analysis). Blue represents follow-up time available in data
after time zero. Black circles represent the end of available data for each person (whether an
event or censoring). Orange circles represent the initiation of a DMARD prescription. Person
A's data are compatible with the MTX monotherapy strategy, and Person B and C's data are
compatible with the MTX+DMARD strategy. The treatment assignment of individuals like
Person D can introduce immortal time bias into analysis, as assigning them all to MTX mono-
therapy artificially inflates the risk estimates made during the grace period, making MTX
monotherapy (possibly incorrectly) appear to be worse than MTX+DMARD.
(TIF)

**S2 Fig. Observability of electronic health records for emulation of target trial to study the
comparative effectiveness of initiating second-line DMARD therapy after methotrexate on
cardiovascular outcomes in rheumatoid arthritis patients, northwestern medicine, January
2000 to June 2020.**
(TIF)

**S3 Fig. Selection of analytic cohort for emulation of target trial to study the comparative
effectiveness of initiating second-line DMARD therapy after methotrexate on cardiovascu-
lar outcomes in rheumatoid arthritis patients, northwestern Medicine, January 2000 to
June 2020.** Abbreviations: DMARD–disease-modifying antirheumatic drug, MACE–major
adverse cardiac event, MTX–methotrexate, RA–rheumatoid arthritis [a]Individuals who initi-
ated a DMARD before time zero were excluded as we could not capture the point in the clini-
cal decision making process when a choice regarding second line therapy was made. [b]For
laboratory values, we imputed missing baseline laboratory data using random-forest based sin-
gle imputation before applying the criteria for inclusion. Laboratory eligibility criteria
included: Platelet>100,000/mm$^3$, estimated glomerular filtration rate>60 mL/min, White
blood cell count>3,000/mm$^3$, Absolute neutrophil count>1200/mm$^3$, Liver trans-
aminases<1.5x upper limit of normal, Hemoglobin>9 g/dL, and Hematocrit>30%.
(TIF)

**S4 Fig. Unadjusted and unweighted survival curves without accounting for immortal
timbe bias.** Note: MTX only - only used methotrexate throughout the grace period, MTX+-
DMARD–added disease-modifying antirheumatic drug to methotrexate at some point during
the grace period.
(TIF)

**S1 File. Supplemental methods and sample R code.**
(DOCX)

**S1 Table. ICD codes for different conditions.**
(DOCX)

**S2 Table. Types of first DMARD started during grace period and average time to starting
DMARD (n = 289), northwestern medicine, January 2000 to June 2020.**
(DOCX)

## Author Contributions

**Conceptualization:** Adovich S. Rivera, Jacob B. Pierce, Matthew J. Feinstein, Lucia C. Petito.

**Data curation:** Anna E. Pawlowski.

**Formal analysis:** Adovich S. Rivera, Lucia C. Petito.

**Methodology:** Adovich S. Rivera, Jacob B. Pierce, Arjun Sinha, Donald M. Lloyd-Jones, Yvonne C. Lee, Matthew J. Feinstein, Lucia C. Petito.

**Resources:** Matthew J. Feinstein.

**Supervision:** Lucia C. Petito.

**Visualization:** Adovich S. Rivera.

**Writing – original draft:** Adovich S. Rivera, Lucia C. Petito.

**Writing – review & editing:** Jacob B. Pierce, Arjun Sinha, Anna E. Pawlowski, Donald M. Lloyd-Jones, Yvonne C. Lee, Matthew J. Feinstein.

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
