## [Decision Letter · Decision Letter 0]

26 Dec 2023

PONE-D-23-36062Designing Target Trials Using Electronic Health Records: A Case Study of Second-Line Disease-modifying Anti-Rheumatic Drugs and Cardiovascular Disease Outcomes in Patients with Rheumatoid ArthritisPLOS ONE

Dear Dr. Rivera,

Thank you for submitting your manuscript to PLOS ONE. After careful consideration, we feel that it has merit but does not fully meet PLOS ONE’s publication criteria as it currently stands. Therefore, we invite you to submit a revised version of the manuscript that addresses the points raised during the review process.

Please be sure to:revise the manuscript according to the reviewers' comments below.Please submit your revised manuscript by Feb 09 2024 11:59PM. If you will need more time than this to complete your revisions, please reply to this message or contact the journal office at plosone@plos.org. Please include the following items when submitting your revised manuscript:A rebuttal letter that responds to each point raised by the academic editor and reviewer(s). You should upload this letter as a separate file labeled 'Response to Reviewers'.A marked-up copy of your manuscript that highlights changes made to the original version. You should upload this as a separate file labeled 'Revised Manuscript with Track Changes'.An unmarked version of your revised paper without tracked changes. You should upload this as a separate file labeled 'Manuscript'.If applicable, we recommend that you deposit your laboratory protocols in protocols.io to enhance the reproducibility of your results. Protocols.io assigns your protocol its own identifier (DOI) so that it can be cited independently in the future. For instructions see: https://journals.plos.org/plosone/s/submission-guidelines#loc-laboratory-protocols. Additionally, PLOS ONE offers an option for publishing peer-reviewed Lab Protocol articles, which describe protocols hosted on protocols.io. Read more information on sharing protocols at https://plos.org/protocols?utm_medium=editorial-email&utm_source=authorletters&utm_campaign=protocols.

We look forward to receiving your revised manuscript.

Kind regards,

Jayeshkumar Patel

Academic Editor

PLOS ONE

Journal Requirements:

"ASR was supported by the American Heart Association Predoctoral Fellowship (825793) for unrelated research. LCP receives funds for unrelated research from Omron Healthcare Co., Ltd. Other authors have no other conflicts to declare. The mentioned organizations had no role in study design, data collection and analysis, decision to publish, or preparation of the manuscript."

Additional Editor Comments:

Dear authors,

Thank you for considering PLOS One to publish your research. Two reviewers and I have completed the review of the manuscript and have a few minor comments for your consideration. Please revise and resubmit the manuscript for further consideration for publication.

Sincerely,

Jayeshkumar Patel

Reviewers' comments:

Reviewer's Responses to Questions

**Comments to the Author**

1. Is the manuscript technically sound, and do the data support the conclusions?

Reviewer #1: Yes

Reviewer #2: Yes

2. Has the statistical analysis been performed appropriately and rigorously?

Reviewer #1: Yes

Reviewer #2: Yes

3. Have the authors made all data underlying the findings in their manuscript fully available?

Reviewer #1: Yes

Reviewer #2: No

4. Is the manuscript presented in an intelligible fashion and written in standard English?

Reviewer #1: Yes

Reviewer #2: Yes

5. Review Comments to the Author

Reviewer #1: Line 95 - The issues that you have laid out here are study design considerations and can occur even while studies are using EHR data...though EHR data may have more nuanced information, biases may occur through study design and not necessarily the data source. Struggling a bit to understand the goal/objective of this study.

Table 2, under eligibility criteria for TT emulation - what is the pre-defined baseline period for exclusion of prior serious comorbidities?

Table 2, under treatment strategies for TT protocol - is there a specific reason for considering only those patients who initiate a DMARD within 24months? you may lose out on patients who for example may initiate a DMARD post 24months.

Table 2, under treatment strategies for TT emulation - I am a bit unclear on how randomization is conducted here...can you please elaborate?

Line 215, "As we did not have cause of death recorded in the EHR, we instead used death from all causes in our definition of MACE"...this is an important study limitation and needs to be added in the limitations section. Also, specify how it may affect the current results.

Reviewer #2: Dear Authors,

This was a well presented paper. on an important topic. I did not have any major concerns. Some suggestions/

clarifications noted below:

Lines 95-96: "Meta-analyses of RCTs only did not conclude that the addition of DMARDs reduced CVD risk in RA patients.....". Does this mean the meta analysis DID conclude that DMARDs increased CVD risk? Can this be re-written?

Lines 248-249: "Other examples of protocols we could have specified are:....." Can you please clarify in the manuscript if this means that the authors could have specified but did not?

Line 296: "we excluded hydroxychloroquine (HCQ) as a DMARD option..." Was there a reason for specifically excluding HCQ?

6. PLOS authors have the option to publish the peer review history of their article (what does this mean?). If published, this will include your full peer review and any attached files.

**Do you want your identity to be public for this peer review?** For information about this choice, including consent withdrawal, please see our Privacy Policy.

Reviewer #1: No

Reviewer #2: No

---

## [Author Response · Author response to Decision Letter 0]

8 Feb 2024

We submitted a properly formatted response as a separate word document.

Reviewer #1: 

Comment 1: Line 95 - The issues that you have laid out here are study design considerations and can occur even while studies are using EHR data...though EHR data may have more nuanced information, biases may occur through study design and not necessarily the data source. Struggling a bit to understand the goal/objective of this study.

Response: We agree that several issues with EHR data also occur with other types of data, the goal of our work is to specifically highlight considerations when using EHR data and the solutions that would likely help address them. We chose to demonstrate this with a case that has not yet been tackled (to our knowledge) that is mostly possible with EHR (instead of research cohort data).

Changes in text (page 4, lines 73-75): 

“Here, we summarize principles in TT emulation using EHR data and provide additional details about design and implementation to supplement existing guides to TT emulation. Additionally, we provide considerations specific to this research question with the hope that readers will consider this guide when applying the TT approach to their own work.”

Comment 2: Table 2, under eligibility criteria for TT emulation - what is the pre-defined baseline period for exclusion of prior serious comorbidities?

Response: For this emulation, we assessed comorbidities using all the data available on the patient in the EHR up to time zero. We added that detail in Table 2.

Changes in text (page 8, Table 2, row on Eligibility Criteria):: 

“Physician confirmation of no prior history of serious cardiovascular disease including … or cancer excluding nonmelanoma skin cancer prior to time zero.”

Comment 3: Table 2, under treatment strategies for TT protocol - is there a specific reason for considering only those patients who initiate a DMARD within 24months? you may lose out on patients who for example may initiate a DMARD post 24months.

Response: The reviewer is correct that choosing to use a 24-month grace period leads us to “lose out on patients” (n=77) — in this case, the people who initiate DMARD after 24 months are classified as MTX monotherapy only. Indeed, the extreme case of including everyone who ever started DMARD, while possible to execute in statistical software, would produce results that are contaminated by immortal time bias (https://doi.org/10.1016/j.jclinepi.2016.04.014). 

While we could extend the grace period, capturing more people in the DMARD plus MTX group, we lose time to do post-intervention follow-up as our data captures a finite observation duration. We chose 24 months because it allows us to include most people who initiated a DMARD during follow-up while balancing the need to specify a clear intervention that reflects clinical practice and can be implemented in real life (mimicking a pragmatic trial). Given this choice, we also implemented a sensitivity analysis that shortened the grace period, and found that our results did not materially change. As seen in this snippet from Table 4:

Table 4. Hazard ratios, risk differences, and restricted mean survival times for 5-year risk of MACE comparing Methotrexate Monotherapy and Addition of Second-Line DMARD Therapy, Northwestern Medicine, January 2000–June 2020.

Analysis† Marginal HR* Risk Difference at month 60* RMST at month 60*

Main 0.717

(0.709 1.228) -1.47 

(-4.74, 1.95)

 0.573

(-0.751, 1.807)

Sensitivity analyses

12-month grace period 0.723

(0.537, 1.270) -2.1

(-6.86, 2.54) 0.778

(-0.945, 2.508)

Changes in text: none

Comment 4: Table 2, under treatment strategies for TT emulation - I am a bit unclear on how randomization is conducted here...can you please elaborate?

Response: We would like to clarify that in this observational study, we do not perform randomization. Rather, we hope to emulate the data structure achieved in an RCT. Specifically, exchangeability of treatment groups, which in RCTs is a consequence of random treatment assignment, and in this target trial is the assumption of random assignment within strata of baseline covariates (i.e., baseline covariate adjustment). This is described in the “Assignment procedures” row of Table 2.

Changes in text: none

Comment 5: Line 215, "As we did not have cause of death recorded in the EHR, we instead used death from all causes in our definition of MACE"...this is an important study limitation and needs to be added in the limitations section. Also, specify how it may affect the current results.

Response: We appreciate this comment by the reviewer and added this in the limitations.

Chang in text (page 23, lines 381-384): 

“Third, our definition of MACE used all-cause death instead of CVD-specific death. While this definition is consistent with some other studies, these results may not be directly comparable to those from studies whose MACE definition included CVD-specific death.[45]”

Reviewer #2: 

Comment 6: This was a well presented paper. on an important topic. I did not have any major concerns. Some suggestions/clarifications noted below:

Response: Thank you!

Comment 7: Lines 95-96: "Meta-analyses of RCTs only did not conclude that the addition of DMARDs reduced CVD risk in RA patients.....". Does this mean the meta analysis DID conclude that DMARDs increased CVD risk? Can this be re-written?

Response: We apologize for the confusion and have edited the sentence to improve clarity.

Changes in text (in yellow):

“Meta-analyses including only RCTs concluded that the addition of DMARDs did not reduce CVD risk in RA patients, while another meta-analysis that included both RCTs and observational studies suggested that adding DMARDs provided some benefit.[23,24]”

Comment 8: Lines 248-249: "Other examples of protocols we could have specified are:....." Can you please clarify in the manuscript if this means that the authors could have specified but did not?

Response: The reviewer is correct that these are examples that we did not implement in the analysis. We added a phrase to clarify this point.

Changes in text (page 4, lines 83-85): 

“Other examples of protocols we could have specified (but did not implement here) are: requiring individuals to….”

Comment 9: Line 296: "we excluded hydroxychloroquine (HCQ) as a DMARD option..." Was there a reason for specifically excluding HCQ?

Response: This sensitivity analysis was done to emulate previously conducted RCTs where they studied the efficacy of adding DMARDs to MTX compared with MTX alone. In these trials, HCQ was not considered step-up therapy, but were included as part of the specified protocol.

Changes in text (page 17, lines 285-286): 

“Exclusion of HCQ was done to emulate some previously conducted RCTs where HCQ was allowed as a concurrent therapy to MTX but was not counted as a step-up DMARD.[38]”

---

## [Decision Letter · Decision Letter 1]

1 May 2024

PONE-D-23-36062R1Designing Target Trials Using Electronic Health Records: A Case Study of Second-Line Disease-modifying Anti-Rheumatic Drugs and Cardiovascular Disease Outcomes in Patients with Rheumatoid ArthritisPLOS ONE

Dear Dr. Rivera,

Thank you for submitting your manuscript to PLOS ONE. After careful consideration, we feel that it has merit but does not fully meet PLOS ONE’s publication criteria as it currently stands. Therefore, we invite you to submit a revised version of the manuscript that addresses the points raised during the review process.

The manuscript has been evaluated by three reviewers, and their comments are available below.Although reviewers 1 and 2 are happy with the revisions made following their previous comments, a third reviewer has also examined the analytic aspects of your manuscript and has raised some concerns - please see their comments below.Could you please revise the manuscript to carefully address the concerns raised?

We look forward to receiving your revised manuscript.

Kind regards,

Steve Zimmerman, PhD

Senior Editor, PLOS ONE

Journal Requirements:

Reviewers' comments:

Reviewer's Responses to Questions

**Comments to the Author**

1. If the authors have adequately addressed your comments raised in a previous round of review and you feel that this manuscript is now acceptable for publication, you may indicate that here to bypass the “Comments to the Author” section, enter your conflict of interest statement in the “Confidential to Editor” section, and submit your "Accept" recommendation.

Reviewer #1: All comments have been addressed

Reviewer #2: All comments have been addressed

Reviewer #3: (No Response)

2. Is the manuscript technically sound, and do the data support the conclusions?

Reviewer #1: Yes

Reviewer #2: Yes

Reviewer #3: Yes

3. Has the statistical analysis been performed appropriately and rigorously? 

Reviewer #1: Yes

Reviewer #2: Yes

Reviewer #3: No

4. Have the authors made all data underlying the findings in their manuscript fully available?

Reviewer #1: Yes

Reviewer #2: Yes

Reviewer #3: No

5. Is the manuscript presented in an intelligible fashion and written in standard English?

Reviewer #1: Yes

Reviewer #2: Yes

Reviewer #3: Yes

6. Review Comments to the Author

Reviewer #1: (No Response)

Reviewer #2: Thank you again for the opportunity to review. No further comments from me. Thank you for addressing my comments.

Reviewer #3: My primary concern is whether the proposed model has been validated, such as through simulation. What are the benefits gained compared to other approaches? Specifically, I am wondering how much bias will be reduced compared to a simple non-weighted model.

A series of sensitivity analyses were conducted, revealing that some estimates closely align while others do not. Has a model selection procedure been employed? What conclusions can be drawn from these diverse results, and how should they be interpreted? What recommendations emerge from this analysis?

Table 3. Why the two groups were presented at 24m? Why not at baseline for baseline characteristics?

Table 4. Can the comorbidity status be used in the weighting and in the model twice time?

7. PLOS authors have the option to publish the peer review history of their article (what does this mean?). If published, this will include your full peer review and any attached files.

Reviewer #1: No

Reviewer #2: **Yes: **Pragya Rai

Reviewer #3: No

---

## [Author Response · Author response to Decision Letter 1]

11 May 2024

Please see below the reviewer comments and corresponding response. A properly formatted detailed responses with changes in the manuscript has been submitted as well.

--------

Comment 1: My primary concern is whether the proposed model has been validated, such as through simulation. What are the benefits gained compared to other approaches? Specifically, I am wondering how much bias will be reduced compared to a simple non-weighted model.

Response: We have edited the introduction, methods, and discussion section of the paper to stress that the advantages of this approach does not lie on the statistical analysis alone. Specifically, the TT approach ensures that the researcher properly specifies the intervention under study. For our research question, we are specifically concerned about the inappropriate practice of comparing ever vs never DMARD users when faced with a scenario where treatment can occur at multiple time points. We highlight Danaei 2011’s (doi: 10.1177/0962280211403603) point that this comparison leads to “a gamble in which the investigators bet that the amount of selection bias introduced is less than the amount of confounding eliminated.” Instead of taking this gamble, our elaborate approach allows handling of both selection and confounding.

To answer the concerns of the validity of the statistical estimation approach (weighted pooled logistic regression), we add details about related simulations which assessed the performance of statistical estimation approaches that can be used in this setting. We think simulation is outside the scope of this paper and the references adequately support the validity of the model we used in the paper. We think a non-weighted model is inconsistent with the various biases we needed to account for especially with the introduction of a grace period. However, we have conducted a naïve analysis which compares ever vs never users to see how much selection bias may have influenced the findings.

Comment 2: A series of sensitivity analyses were conducted, revealing that some estimates closely align while others do not. Has a model selection procedure been employed? What conclusions can be drawn from these diverse results, and how should they be interpreted? What recommendations emerge from this analysis?

Response: We revised the paper to clarify that we conducted theory-based variable selection with the assistance of directed acyclic graphs. This is the recommended approach instead of using data driven covariate selection approaches (e.g., lasso regression).(see Staerk and Mayr 2024, doi.org/10.1093/aje/kwad193) We have also modified the discussion to state more clearly what can be concluded from the sensitivity analysis and the implications of this work for research practice.

Comment 3: Table 3. Why the two groups were presented at 24m? Why not at baseline for baseline characteristics?

Response: We appreciate this concern. We do present baseline characteristics for the overall sample in Table 3. However, at baseline treatment assignment is not finalized – individuals may be compatible with more than one treatment arm as we are comparing treatment initiation by the end of the grace period, which ends after study baseline. So, instead we conducted the comparison of covariates at the end of the grace period since it is only at this point where the exposure assignment is firmly assigned. 

Comment 4: Table 4. Can the comorbidity status be used in the weighting and in the model twice time?

Response: To answer the reviewer’s query, including the same covariate in both the weighting and outcome model is accepted practice. It is one way to address any residual imbalance that is incompletely addressed by weighting. We added this detail and the appropriate citation in the text.

---

## [Decision Letter · Decision Letter 2]

31 May 2024

Designing Target Trials Using Electronic Health Records: A Case Study of Second-Line Disease-modifying Anti-Rheumatic Drugs and Cardiovascular Disease Outcomes in Patients with Rheumatoid Arthritis

PONE-D-23-36062R2

Dear Dr. Rivera,

We’re pleased to inform you that your manuscript has been judged scientifically suitable for publication and will be formally accepted for publication once it meets all outstanding technical requirements.

Kind regards,

Steve Zimmerman, PhD

Senior Editor, PLOS ONE

Additional Editor Comments (optional):

Reviewers' comments:

Reviewer's Responses to Questions

**Comments to the Author**

1. If the authors have adequately addressed your comments raised in a previous round of review and you feel that this manuscript is now acceptable for publication, you may indicate that here to bypass the “Comments to the Author” section, enter your conflict of interest statement in the “Confidential to Editor” section, and submit your "Accept" recommendation.

Reviewer #3: All comments have been addressed

2. Is the manuscript technically sound, and do the data support the conclusions?

Reviewer #3: (No Response)

3. Has the statistical analysis been performed appropriately and rigorously? 

Reviewer #3: (No Response)

4. Have the authors made all data underlying the findings in their manuscript fully available?

Reviewer #3: (No Response)

5. Is the manuscript presented in an intelligible fashion and written in standard English?

Reviewer #3: (No Response)

6. Review Comments to the Author

Reviewer #3: All my concerns are addressed.

The statistics are acceptable now.

7. PLOS authors have the option to publish the peer review history of their article (what does this mean?). If published, this will include your full peer review and any attached files.

Reviewer #3: No

---

## [Editor Report · Acceptance letter]

5 Jun 2024

PONE-D-23-36062R2 

PLOS ONE

Dear Dr. Rivera, 

I'm pleased to inform you that your manuscript has been deemed suitable for publication in PLOS ONE. Congratulations! Your manuscript is now being handed over to our production team.

Kind regards, 

on behalf of

Dr Steve Zimmerman 

Staff Editor

PLOS ONE